# Cardiometabolic Risk Factor in Obese and Normal Weight Individuals in Community Dwelling Men

**DOI:** 10.3390/ijerph17238925

**Published:** 2020-11-30

**Authors:** Hyunsoo Kim, Kijeong Kim, Sohee Shin

**Affiliations:** 1Department of Sports Science, Seoul National University of Science and Technology, 232 Gongneung-ro, Nowon-gu, Seoul 01811, Korea; hskim@seoultech.ac.kr; 2School of Sport and Exercise Science, University of Ulsan, 93 Daehak-ro, Nam-gu, Ulsan 44610, Korea; kijeongk@ulsan.ac.kr

**Keywords:** body mass index, waist circumference, cardiometabolic risk factors

## Abstract

The aim of this study was to investigate the cardiometabolic risk factors (CRFs) in community dwelling men based on a combination of body mass index (BMI) and waist circumference (WC). This cross-sectional study was based on 867 males between the ages of 20 and 71 years. Subjects were categorized into 4 groups by BMI and WC (Group 1, BMI < 25 kg/m^2^ and WC < 90 cm; Group 2, BMI < 25 kg/m^2^ and WC > 90 cm; Group 3, BMI > 25 kg/m^2^ and WC < 90 cm; and Group 4 BMI > 25 kg/m^2^ and WC > 90 cm). The proportion of subjects with a normal weight with high WC was 3.2%. Among normal weight men with the high range of WC, significantly high Odds ratios (ORs) and 95% CI were found for hypertriglyceridemia (3.8, 1.8–8.2) and high blood glucose (3.2, 1.5–6.9). The probability that the general obesity group (Group 3) had one CRF was around twice that of the reference group (Group 1) (1.9 to 2.1 times), but Group 2 had probability more than 4 times higher (4.3 to 4.6 times). In community dwelling adult men, normal weight with high waist circumference was associated with the highest cardiometabolic risk. In conclusion, follow-up screening of those with high WC may be necessary to detect and prevent cardiometabolic diseases, particularly for men with a normal weight.

## 1. Introduction

Obesity is an important risk factor for cardiometabolic diseases such as diabetes, hypertension, dyslipidemia, and coronary artery disease [1,2]. The WHO and the NIH provide guidelines for classifying obesity based on body mass index (BMI) [1,2], and many epidemiological studies have shown that there is a direct relationship between BMI and morbidity and mortality [3,4]. The diagnosis of obesity is often based on BMI, calculated as weight in kilograms divided by height in meters squared (kg/m^2^). For example, Chang et al. [5] reported that in a population of Korean men, the rate of development of metabolic syndrome was 68% greater among those with BMI > 25 compared to those with BMI between 18.5 and 22.9; when limited to weight-stable subjects the rate of metabolic syndrome in this BMI group increased to more than five times the rate in the lower BMI category.

BMI is a simple and inexpensive measurement method and is widely used as an index for evaluating obesity in epidemiological studies or in the field. However, it does not accurately reflect muscle mass, body fat mass or body fat distribution [6]. For example, people with a large amount of body fat and low weight, such as the elderly, can be evaluated as normal, whereas young people or athletes with low fat and large amounts of muscle are often evaluated as obese [7]. In particular, this method does not accurately assess the risk of diseases associated with obesity [8].

Accumulation of visceral fat is known to increase cardiometabolic risk factors (CRFs) and mortality [9]. Waist circumference (WC) is a simple and valuable anthropometric measure of total and intra-abdominal body fat [10]. WC is often used as a surrogate marker of abdominal fat mass because WC correlates with abdominal fat mass [11] and is associated with cardiometabolic disease risk [12]. Therefore, it is important to examine the proportion of people who are a normal weight but abdominally obese and those who are obese but not abdominally obese among the community dwelling people who are tested for metabolic syndrome at public health centers, and whether there are differences in the CRFs among these groups. In most previous studies with Korean participants, researchers took into account BMI and/or percentage body fat but did not evaluate the combined effects of WC and BMI on CRFs [13,14]. In addition, it is known that men have more abdominal obesity than women and have a higher risk of metabolic disease. Therefore, this study was conducted to assess the characteristics of CRFs in community dwelling men with a high WC and normal BMI.

## 2. Materials and Methods

### 2.1. Participants

Participants were 867 healthy males (45.1 ± 14.2 years old) aged from 20 to 71 years old who had a health checkup at the Nowon-Gu public health center in Seoul. They are residents of Nowon-gu and voluntary participants in free metabolic syndrome tests. They were divided into 4 groups according to BMI and WC. Group 1 is BMI < 25 kg/m^2^, WC < 90 cm, Group 2 is BMI < 25 kg/m^2^, WC ≥ 90 cm, Group 3 is BMI ≥ 25 kg/m^2^, WC < 90 cm, and Group 4 is BMI ≥ 25 kg/m^2^, WC ≥ 90 cm. Group 1, who were not obese by BMI and WC standards, was used as the reference group. This study was conducted using data opened for research purposes at Nowon-gu Health Center. This study was done with reference to the Declaration of Helsinki of 1964, and the subjects who participated in the health checkup agreed that their own health checkup data would be used for the study in an anonymous form.

### 2.2. Measurement

Height and weight were measured using a height/weight meter (BIKI200), without shoes while wearing light clothing. BMI was calculated as weight in kilograms divided by the square of the height in meters. WC was measured in cm at the level of the belly button after a normal expiration with a tape measure once. Body fat (kg), body muscle (kg) and BF% were simultaneously measured using a multi-frequency impedance device (T-scan Plus; Jawon Medical, Seoul, Korea).

### 2.3. Criteria of Obesity and Cardiovascular Disease Risk Factors

In our study, obesity by BMI was defined as a BMI of at least 25 kg/m^2^, which has been recommended for Asians by the World Health Organization [15] although still debatable, in clinical setting and several epidemiologic studies [16]. The present study used the Asia-Pacific BMI classification by the World Health Organization [15]. The cutoff points of WC for abdominal obesity are ≥90 cm for men. This is used to indicate central obesity according to the Asian-specific WC cutoff points of the International Diabetes Federation criteria [17].

Systolic and diastolic blood pressures were measured in the right arm using an automatic manometer (FT-200S; Jawon medical, Kyungsan, Korea) in the sitting position after a 10-min rest period. During the 30 min preceding the measurement, the subjects were required to refrain from smoking or consuming caffeine. Blood samples were drawn from an antecubital vein into Vacutainer tubes containing Ethelene-Diamine-Tetra-Acetic acid (EDTA) in the morning after an overnight fast. Fasting plasma glucose (FPG), total cholesterol (TC), triglyceride (TG), and high-density lipoprotein cholesterol (HDL-C) were measured using an auto-analyzer (Hitachi 7600; Hitachi, Tokyo, Japan). Low-density lipoprotein cholesterol (LDL-C) was calculated by the Friedewald equation (LDL-C = TC − HDL-C − (TG/5)) [18].

The risk factors for cardiometabolic disease were defined as follows [17]:(1)High blood pressure: a systolic blood pressure of at least 130 mm Hg and/or a diastolic blood pressure of at least 85 mm Hg and/or treatment for previously diagnosed hypertension.(2)Hyperglycemia: an FPG of at least 100 mg/dL (≥5.6 mmol/L) and/or treatment for previously diagnosed type 2 diabetes mellitus.(3)Dyslipidemia: a TG of at least 150 mg/dL (≥1.7 mmol/L) and/or an HDL-C less than 40 mg/dL (b1.03 mmol/L) and/or an LDL-C of at least 160 mg/dL (≥4.1 mmol/L) and/or treatment for previously diagnosed dyslipidemia.

### 2.4. Statistical Analysis

The differences among groups in terms of the physical characteristics and CRFs of the participants were analyzed by one-way ANOVA. If there was a significant difference between groups, Tukey’s HSD (honestly significant difference) method was used as a post-hoc comparison. Binomial logistic regression analysis was performed to determine the effect on the presence or absence of CRFs, according to group classification. Odds ratio (OR), 95% CI and significance probability were calculated based on the reference group (Group 1). SPSS ver.23.0 for Window (SPSS, Inc., Chicago, IL, USA) was used for all statistical analysis, and the statistical significance level was set to 5%.

## 3. Results

Table 1 shows the characteristics of participants. Of all subjects, 60% (519 people) were classified into Group 1, whose weight and WC were considered normal. The abdominal obesity group (Group 2), whose BMI is normal but WC is large, and the general obesity group (Group 3), whose WC is normal but are considered obese because of their BMI, had 3% (28 people) and 14% (125 people), respectively. The composite obesity group (Group 4), whose weight and WC were both obese, accounted for 22% (195 people).

The results of the descriptive statistics of CRFs among each group and the differences between the groups are shown in Table 2. All CRFs showed significant differences among the groups. In the obese groups, Group 2 showed higher values than Group 3 in triglyceride. The blood glucose levels were also higher in Group 2 than in Group 3. While, compared to the reference group, Group 3 and Group 4 showed higher values in systolic blood pressure (SBP) and diastolic blood pressure (DBP) and lower values in high-density lipoprotein cholesterol (HDLC). Group 4 showed higher values than the reference group in low-density lipoprotein cholesterol (LDLC) (1 < 4).

Table 3 and Figure 1 show the results of a binary logistic analysis to determine the effect on the presence and number of risk factors for metabolic syndrome by group classification. The probability of triglyceride, LDLC and blood glucose levels in Group 2 compared to the reference group were 3.8 times, 2.7 times, and 3.2 times higher, respectively, and these risk levels are higher than that of Group 4. Group 3, compared to the reference group, had higher probabilities of worsening triglyceride, total cholesterol and hypertension levels: 1.9, 1.5 and 2.1 times higher, respectively. Compared to the reference group, the probability that Group 2 had more than one risk factor was 4.3 times higher, that of having two or more risks was 4.4 times higher, and for three or more it was 4.6 times higher. These risk levels are higher than those of Group 4.

## 4. Discussion

The present study was conducted to examine the extent of abdominal obesity assessed by WC, and the relationship between increased WC and CRFs, such as hypertension, diabetes and dyslipidemia in community dwelling men. The prevalence of cardiometabolic abnormal subjects among normal weight subjects was 10.5%. This is in agreement with data from the 3rd Korea National Health and Nutrition Examination Survey, which showed that 12.7% of subjects with a BMI < 25 kg/m^2^ had metabolic syndrome [16].

Abdominal obesity, assessed by WC, has been found to be a better predictor of all-cause and cardiovascular disease mortality than BMI in some population groups [19,20,21]. In the present study, the abdominal obesity group (Group 2) had more CRFs than the general obesity group (Group 3). The adjusted OR of abdominal obesity with more than one CRF was 4.31 (95% CI 1.28–14.48). Coutinho et al. [18] reported that the mortality rate was the highest in the group with a high WC value in the normal BMI. The reason why abdominally obese people with normal weight in this study have many risk factors is not only a physiological mechanism, but also is because they receive less preventive treatment, such as healthy diet advice and exercise recommendations, than general obesity in the clinical field.

Group 2 and Group 3 showed similar amounts of body fat and percentage body fat. It is known that there is a high probability of having a metabolic disease if the body weight is the same but the muscle is low or the amount of body fat is high [22]. However, in this study, the relative and absolute body fat and muscle mass of the two groups did not show any significant difference. Even if the total amount of fat is the same, a large amount of visceral fat increases the risk of CRFs and mortality [10,11]. WC is a simple and important anthropometric variable that can predict total body fat and abdominal fat. An increase in the WC means that the amount of visceral fat has increased [14]. So, it can be seen that increasing the WC worsens CRFs and health risks such as diabetes, heart disease and mortality [8,22].

In the present study, 41% of persons had >1 MS risk factor, and 12% had ≥2 risk factors in the reference group. Conus et al. [19] reported that the percentage of metabolic unhealthy normal weight was 5–45%. Meigs et al. [23] showed that both heart disease and diabetes were more prevalent in those of a metabolic unhealthy normal weight than in those were considered metabolic healthy obese, confirming that assessing health status by body weight alone is not desirable. Meigs et al. [23] reported that metabolically abnormal normal weight subjects were at a 2- to 3-fold increased risk of cardiovascular disease relative to normal weight subjects without metabolic syndrome. Shin et al. [24] reported that the risk of having two or more CRFs in the abdominal obesity group and general obesity group were 1.8 times and 2.6 times those of the reference group in community dwelling women. However, this study shows that the probability of having two or more CRFs was 4.4 times and 1.9 times that of the reference group in the community dwelling men. In other words, the odds ratio of the abdominal obesity group was higher than that of the general obesity group. These results suggest that there is a gender difference between abdominal obesity groups, and abdominal obesity in males is more likely to carry a risk of cardio-metabolic disease than in females.

In normal-weight men, elevated WC was associated with significantly higher odds of having the CRF and in particular high triglycerides and blood sugar levels. Lemieux et al. [25] proposed that the hyper-triglyceridemic high waist (waist girth ≥90 cm combined with fasting plasma TG levels ≥2.0 mmol/L) was predictive of a very high probability for men to be characterized by the simultaneous presence of some cardiometabolic risk markers. It has also been suggested that the hyper-triglyceridemic high waist could be helpful in the assessment of risk of coronary artery disease and type 2 diabetes mellitus [25,26]. Hanley et al. [27] suggested that impaired glucose tolerance in abdominal obese people with high triglycerides is the strongest predictor of diabetes incidence. Korea is very fast in enacting socioeconomic and political changes. In transitional countries where socioeconomic and political changes occur rapidly, WC increases faster than body weight [28]. Changes in body shape that increase WC will increase the risk of circulatory and metabolic diseases, will impose great burdens on Korean society in the future. Therefore, men with high triglycerides and high WC should be classified as high-risk groups to prevent deterioration.

In the present study, 66% of those with one or more CRFs, and 16% of those with three or more, were in the reference group. Likewise, the reason for metabolic obesity, even though their weight was normal, was due to abnormalities in the amount and distribution of fat, insulin resistance, and fat metabolism, but another cause was the lack of physical activity [20,21]. The rate of energy intake from carbohydrates and fat was not associated with the risk of metabolic obesity in normal weight people [29]. Conus et al. [19] and Lee [16] mentioned that the most significant cause of being metabolic unhealthy in normal weight was a lack of physical activity. Therefore, even if the body weight is within the normal range, people with CRFs should be screened carefully for coronary artery disease, and lifestyle improvement, specifically exercise therapy, should be considered. However, we did not investigate the amount of physical activity done by the participants, and it is not known whether exercise or physical fitness influenced the outcome. In the future, it is necessary to investigate lifestyle habits, especially physical activity and physical fitness, including aerobic capacity.

The first limitation of this study is that it is difficult to generalize because it cannot represent the general population, as it used data from only one health center. Second, factors that may affect the study results, such as drug use, smoking, physical activity and alcohol, all of which influence CRFs, were not sufficiently examined.

## 5. Conclusions

In conclusion, among adult males living in a community, those with high WC had more cardiometabolic risk factors than those with general obesity. Therefore, people with normal weight but high WC should be identified early to inform individuals of the risk of abdominal obesity, and exercise and meal prescriptions, and education should be provided to reduce WC, as well.

## Figures and Tables

**Figure 1 ijerph-17-08925-f001:**
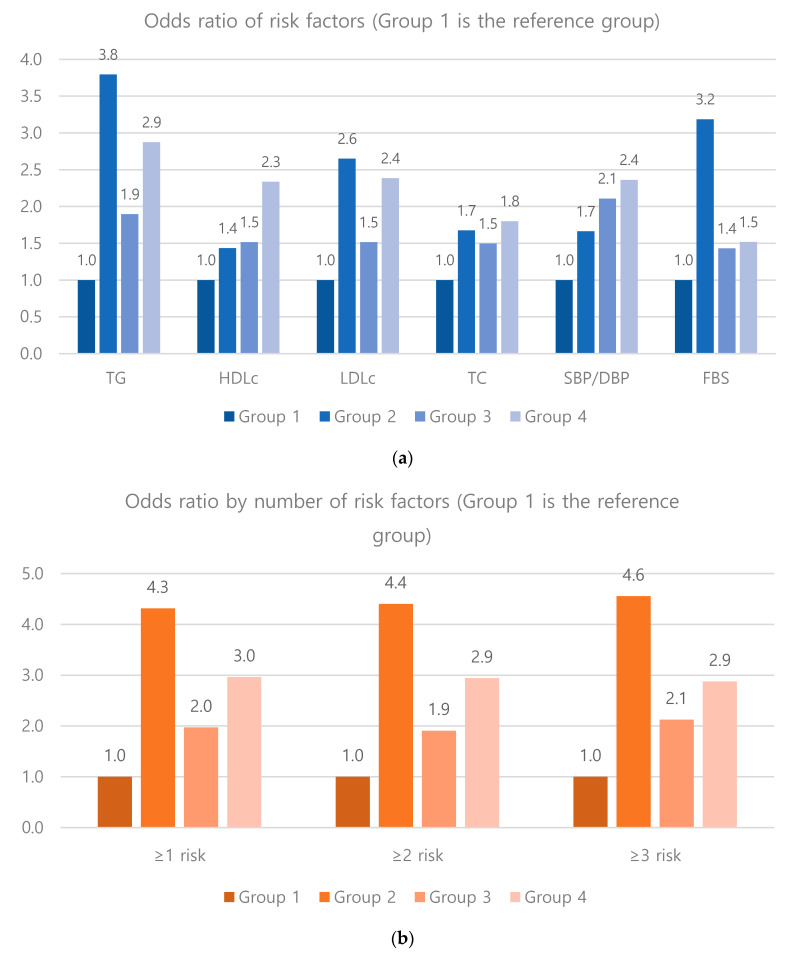
Probability of risk of a problem occurring in metabolic factors. Odds ratios indicates how many times each group has the risks are compared to Group 1. Group; 1: BMI < 25, WC < 90, 2: BMI < 25, WC ≥ 90, 3: BMI ≥ 25, WC < 90, 4: BMI ≥ 25, WC ≥ 90.

**Table 1 ijerph-17-08925-t001:** The physical characteristics for the four groups.

Parameters	Group	n	Mean	SD	One-Way ANOVA		
					F-Value		Tukey’ HSD
Age (yr)	1	519	45.2	14.6	2.0		
	2	28	50.6	12.1			
	3	125	45.5	13.7			
	4	195	43.8	13.5			
Height (cm)	1	519	170.8	6.4	11.3	*	1, 3 < 2, 4
	2	28	174.3	5.9			
	3	125	169.8	6.6			
	4	195	173.1	5.8			
Weight (kg)	1	519	65.1	6.8	324.6	*	1 <2, 3 < 4
	2	28	73.3	4.7			
	3	125	75.4	6.0			
	4	195	84.0	9.7			
BMI	1	519	22.3	1.7	513.9	*	1 < 2 < 3 < 4
	2	28	24.1	0.7			
	3	125	26.1	1.0			
	4	195	28.0	2.5			
WC (cm)	1	519	81.2	5.4	473.5	*	1 < 3 < 2 < 4
	2	28	92.6	1.6			
	3	125	87.2	2.4			
	4	195	96.9	5.6			
SLM (kg)	1	519	48.3	5.6	99.6	*	1 < 2, 3 < 4
	2	28	51.6	3.9			
	3	125	52.6	5.5			
	4	195	56.2	6.0			
BFM (kg)	1	519	12.8	3.3	417.5	*	1 < 2, 3 < 4
	2	28	17.6	1.2			
	3	125	18.6	2.6			
	4	195	22.9	4.7			
PBF (%)	1	519	19.5	4.1	220.2	*	1 < 2, 3 < 4
	2	28	24.0	1.5			
	3	125	24.6	3.3			
	4	195	27.1	3.3			

Note. * *p* < 0.05, Group; 1: BMI < 25, WC < 90, 2: BMI < 25, WC ≥ 90, 3: BMI ≥ 25, WC < 90, 4: BMI ≥ 25, WC ≥ 90, BMI, body mass index; WC, waist circumference; SLM, soft lean mass; BFM, body fat mass; PBF, percent body fat.

**Table 2 ijerph-17-08925-t002:** The mean levels of cardiometabolic risk factors by group.

Parameters	Group	Descriptive Statistics		One-Way ANOVA		Post Hoc.
		Mean	SD	F-Value		Tukey’HSD
TG (mg/dL)	1	128.0	81.7	24.84	*	1 < 2, 3, 4
	2	253.0	299.9			3 < 2
	3	177.6	144.0			
	4	208.2	177.3			
HDLC (mg/dL)	1	51.0	12.1	17.88	*	3, 4 < 1
	2	46.4	11.4			
	3	45.9	9.0			
	4	45.0	9.4			
LDLC (mg/dL)	1	121.8	30.8	10.48	*	1 < 4
	2	134.4	35.2			
	3	128.5	31.7			
	4	136.0	32.2			
TC (mg/dL)	1	189.5	36.1	8.12	*	1 < 4
	2	203.9	39.0			
	3	196.5	37.3			
	4	204.0	39.6			
SBP (mmHg)	1	124.3	10.9	17.94	*	1 < 3, 4
	2	127.1	9.1			
	3	128.0	9.5			
	4	130.9	12.0			
DBP (mmHg)	1	76.5	8.6	10.67	*	1 < 3, 4
	2	78.0	8.0			
	3	79.0	7.7			
	4	80.5	9.9			
FBS (mg/dL)	1	94.3	16.2	8.17	*	1, 3 < 2, 4
	2	112.3	55.8			
	3	95.2	11.5			
	4	100.8	34.4			

Note. * *p* < 0.05, Group; 1: BMI < 25, WC < 90, 2: BMI < 25, WC ≥ 90, 3: BMI ≥ 25, WC < 90, 4: BMI ≥ 25, WC ≥ 90, TG, triglyceride; HDLC, high-density lipoprotein cholesterol; LDLC, low-density lipoprotein cholesterol; TC, Total cholesterol; SBP, systolic blood pressure; DBP, diastolic blood pressure, FBS, fasting blood sugar.

**Table 3 ijerph-17-08925-t003:** Logistic regression analysis on cardiometabolic risk factors.

Parameters	Group	Frequency		Odds Ratio	95% CI	
		n	(%)		Lower	Upper
TG (≥150 mg/dL)	1	135	26%	1.00	(Reference)	
	2	16	57%	3.79 *	1.75	8.22
	3	50	40%	1.90 *	1.26	2.85
	4	98	50%	2.87 *	2.04	4.05
HDLC (<40 mg/dL)	1	83	16%	1.00	(Reference)	
	2	6	21%	1.43	0.56	3.64
	3	28	22%	1.52	0.94	2.45
	4	60	31%	2.33 *	1.59	3.43
LDLC (≥160 mg/dL)	1	58	11%	1.00	(Reference)	
	2	7	25%	2.65 *	1.08	6.50
	3	20	16%	1.51	0.87	2.63
	4	45	23%	2.38 *	1.55	3.67
TC (≥220 mg/dL)	1	194	37%	1.00	(Reference)	
	2	14	50%	1.68	0.78	3.59
	3	59	47%	1.50 *	1.01	2.22
	4	101	52%	1.80 *	1.29	2.51
SBP/DBP (≥85/130 mmHg)	1	60	12%	1.00	(Reference)	
	2	5	18%	1.66	0.61	4.54
	3	27	22%	2.11 *	1.27	3.49
	4	46	24%	2.36 *	1.54	3.62
FBS (≥100 mg/dl)	1	111	21%	1.00	(Reference)	
	2	13	46%	3.19 *	1.47	6.89
	3	35	28%	1.43	0.92	2.23
	4	57	29%	1.52 *	1.05	2.20
1 or more risks	1	342	66%	1.00	(Reference)	
	2	25	89%	4.31 *	1.28	14.48
	3	99	79%	1.97 *	1.23	3.15
	4	166	85%	2.96 *	1.92	4.57
2 or more risks	1	188	36%	1.00	(Reference)	
	2	20	71%	4.40 *	1.90	10.19
	3	65	52%	1.91 *	1.29	2.83
	4	122	63%	2.94 *	2.09	4.14
3 or more risks	1	83	16%	1.00	(Reference)	
	2	13	46%	4.55 *	2.09	9.92
	3	36	29%	2.12 *	1.35	3.34
	4	69	35%	2.88 *	1.98	4.19

Note. * *p* < 0.05, Group; 1: BMI < 25, WC < 90, 2: BMI < 25, WC ≥ 90, 3: BMI ≥ 25, WC < 90, 4: BMI ≥ 25, WC ≥ 90, TG, triglyceride; HDLC, high-density lipoprotein cholesterol; LDLC, low-density lipoprotein cholesterol; TC, Total cholesterol; SBP, systolic blood pressure; DBP, diastolic blood pressure, FBS, fasting blood sugar, Frequency: The number of people with a value higher than the cut-off of each parameters (HDLC was based on a lower value).

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
