# Peer review of "Cardiometabolic Risk Factor in Obese and Normal Weight Individuals in Community Dwelling Men"

_ijerph, 2020, doi:10.3390/ijerph17238925_

Round 1

Reviewer 1 Report

I have some questions and comments.

  1. Abstract: Please present results in OR (95% confidence interval).
  2. L48: How did you choose the participants? Were the participants healthy when they went to have a health checkup at the clinic.
  3. L64: If a participant was taking antihypertension medications and his blood pressures were in normal range, was he hypertensive?
  4. Table 1: What did the “*” indicate? What was “1<2”? if 1<2 meant group 1 mean was less than group 2 mean, was the difference statistically significant? Did you test if mean age was different among the four groups?
  5. Participants in group 4 had significantly higher soft lean mass. How do you interpret it?
  6. Figure 1: Please choose different type of figures and make them look better.
  7. Did you do any multivariate analysis? Why not?
  8. Why did you focus on men in this study?

Reviewer 2 Report

Abstract

Error (]

Introduction

This section does not present information on what obesity is and what is considered normal weight by the scientific community, the calculation of BMI.

Likewise, no information is presented on the men who live in the community. This section needs a clearer definition

No previous studies on this topic.

Methods

Participants.

Is it possible that some subject had a BMI <25kg / m2 and WC ≥ 90cm, could be in group 1 or 2? What number of subjects does each group contain? Are there differences in the age of each of these groups? These differences without significant? It is quite confusing.

Measurement

Instruments to measure height and weight?

Body composition has been analyzed or used in the study. Measurement protocol?

Lines 61-66. It is not method. Could be added to the introduction

How was hypertriglyceridemia and high blood glucose measured?

How triglycerides were measured; high-density lipoprotein cholesterol; low-density lipoprotein cholesterol; Total cholesterol; systolic blood pressure; diastolic blood pressure, fasting blood sugar?

Statistic analysis.

They must explain that they have performed the Tukey'HSD

Research ethics committee. Data protection policy?

The methods do not meet the stated objective.

Results

Table 1: These SLM data, soft lean mass; BFM, body fat mass; PBF, body fat percentage are not anthropometric.

Figure 1: 1, 2, 3, 4. What? Risks? groups?

Resultuado poorly exposed.

Clarify tables and figures

Round 2

Reviewer 1 Report

The revised manuscript looks better. I have no more questions or comments.

Author Response

We are grateful to reviewers for the critical comments and useful suggestions that have helped us to improve our paper considerably. Thank you so much.

Reviewer 2 Report

Dear Authors.

Was the research done in accordance with the Declaration of Helsinki of 1964 and was it approved by a Local Ethics Committee?

Author Response

Thank you for your comments. We have revised ‘participants’ of the method according your comments. Data collection for this study was conducted before the establishment of Local Ethics Committee, however, this study was conducted using data opened for research purposes at Nowon-gu Health Center. This study was done with reference to the Declaration of Helsinki of 1964. The subjects who participated in the health checkup agreed that their own health checkup data would be used for the study in an anonymous form. We kindly ask for your understanding.